# How the Intrinsically Disordered N-Terminus of Cancer/Testis Antigen MAGEA10 Is Responsible for Its Expression, Nuclear Localisation and Aberrant Migration

**DOI:** 10.3390/biom13121704

**Published:** 2023-11-24

**Authors:** Anneli Samel, Fred Väärtnõu, Lisbeth Verk, Kristiina Kurg, Margit Mutso, Reet Kurg

**Affiliations:** Institute of Technology, University of Tartu, 50411 Tartu, Estonia; anneli.samel@ut.ee (A.S.); fred.vaartnou@ut.ee (F.V.); lisbeth.verk@gmail.com (L.V.); kristiina.kurg@ut.ee (K.K.); margit.mutso@ut.ee (M.M.)

**Keywords:** cancer/testis antigens, MAGEA10, intrinsically disordered proteins, nuclear localization, protein stability

## Abstract

Melanoma-associated antigen A (MAGEA) subfamily proteins are normally expressed in testis and/or placenta. However, aberrant expression is detected in the tumour cells of multiple types of human cancer. MAGEA expression is mainly observed in cancers that have acquired malignant phenotypes, invasiveness and metastasis, and the expression of MAGEA family proteins has been linked to poor prognosis in cancer patients. All MAGE proteins share the common MAGE homology domain (MHD) which encompasses up to 70% of the protein; however, the areas flanking the MHD region vary between family members and are poorly conserved. To investigate the molecular basis of MAGEA10 expression and anomalous mobility in gel, deletion and point-mutation, analyses of the MAGEA10 protein were performed. Our data show that the intrinsically disordered N-terminal domain and, specifically, the first seven amino acids containing a unique linear motif, PRAPKR, are responsible for its expression, aberrant migration in SDS-PAGE and nuclear localisation. The aberrant migration in gel and nuclear localisation are not related to each other. Hiding the N-terminus with an epitope tag strongly affected its mobility in gel and expression in cells. Our results suggest that the intrinsically disordered domains flanking the MHD determine the unique properties of individual MAGEA proteins.

## 1. Introduction

Cancer/testis antigens (CTAs) comprise a family of tumour-associated antigens (TAAs) that are aberrantly expressed in malignant tumours of different histological origin but are silenced in most normal tissues, with the exception of the germ line and, in some cases, the placenta and brain [1]. With such a limited expression pattern and the fact that many of the gene products are immunogenic [2,3], CTAs have gathered attention as possible targets for cancer immunotherapy [4].

MAGEA (melanoma associated antigen A) subfamily proteins were the first TAAs identified at the molecular level [5]. They are recognized by cytotoxic T lymphocytes and evoke strong T cell reactivity against autologous tumour cells in culture [1]. *MAGEA* is a subfamily of 12 genes (*MAGEA1* to *-A12*) located in the q28 region of the X chromosome [6]. In normal tissues, *MAGEA* expression has only been detected in the immature cells of the testis or placenta, such as spermatogonia and trophoblasts [6], and in the embryonic nervous system [7], suggesting that *MAGEAs* may function in germ cell and early neuronal development. Several types of malignant tumours also exhibit *MAGEA* expression, most likely caused by genome-wide epigenetic reprogramming occurring during tumorigenesis [8,9]. *MAGEA* expression in cancer has been linked to tumour cell proliferation [10,11], malignancy [12], invasiveness [11], metastasizing [11], and poor patient prognosis [13]. MAGEA proteins are some of the most immunogenic and frequently expressed members of the CTAs and are thus considered excellent candidates for cancer theranostics [14].

All MAGE proteins share a conserved region spanning approximately 170 amino acids, termed the MAGE homology domain (MHD). In the MAGEA subfamily, the MHD is 70% conserved at the amino acid level [15]. Despite this, MAGE proteins are structurally dynamic and therefore have many possibilities for interactions with different partners [16]. The majority of CTAs, specifically those located on the X chromosome, are intrinsically disordered proteins (IDPs), meaning they lack rigid 3D structures in localised regions or along their entire length under physiological conditions [17]. With all of that in mind, MAGEA proteins could have many functions in both cancerous and normal cells. MAGEA proteins are known to not only interact with RING E3 ubiquitin ligases [15], modulate p53 function [18], apoptosis [19] and cell growth [20] but also regulate transcription [21].

MAGEA10 is the most immunogenic protein of the MAGEA subfamily [22,23]. However, the biological, cellular and molecular functions of MAGEA10 still remain poorly understood. MAGEA10 is a nuclear protein with a length of 369 aa and a computational molecular weight of about 42 kDa; however, in sodium dodecyl sulphate–polyacrylamide gel electrophoresis (SDS-PAGE), it migrates approximately as a 70 kDa protein [24,25]. In this current study, we performed deletion and point-mutation analyses of the MAGEA10 protein to investigate the molecular basis of MAGEA10 expression and anomalous mobility in gel. Our data show that the intrinsically disordered N-terminal domain and, specifically, the first seven amino acids containing linear motif PRAPKR are responsible for its expression, its aberrant migration in SDS-PAGE and its nuclear localisation in mammalian cells.

## 2. Materials and Methods

### 2.1. Plasmids

The pQM-MAGEA10 plasmid expresses the *MAGEA10* coding sequence fused in-frame with C-terminal E2Tag epitope under the control of the *CMV* promoter [26]. pQM-MAGEA10-EGFP was constructed by cloning the *MAGEA10* coding sequence into a pEGFP-N1 vector in-frame with the *EGFP*-coding sequence. Truncated versions of MAGEA10 were generated by PCR and cloned into pQM-CMV-E2Tag vector (Icosagen; Tartu, Estonia). The list of primers used (Microsynth AG; Balgach, Switzerland) is shown in Appendix A. To construct *MAGEA10* deletion mutants fused in-frame with *EGFP*, the coding sequence of *TRMT112* in the TRMT112-EGFP plasmid [27] was substituted with *MAGEA10* mutant sequences via restriction with Acc65I and MunI. MAGEA10 amino acid substitution mutants were generated by PCR using the M-PIPE (polymerase incomplete primer extension mutagenesis) method [28] with primers depicted in Appendix A. All of the sequences were verified by sequencing.

### 2.2. Cells and Transfections

Cop5-EBNA mouse fibroblast cells [26] were cultured at 37 °C in IMDM medium (CORNING; Manassas, VA, USA) supplemented with 10% foetal calf serum (PAN-Biotech; Aidenbach, Germany), penicillin (100 U/mL) and streptomycin (100 ng/mL, CORNING; Manassas, VA, USA). The cells were transfected with plasmids coding for different MAGEA10 coding region variants via electroporation at 230 V and 975 μF on the GenePulser Xcell™ (Bio-Rad Laboratories, Hercules, CA, USA), as described in Kurg et al. [26]. Plasmids without the MAGEA10 insert were used as a negative control. For subsequent extracellular vesicle (EV) isolation, the cells were cultured in an IMDM medium supplemented with 5% exosome-free foetal calf serum (PAN-biotech, ultracentrifuged at 100,000× *g* for 4 h and filtered with a 0.22 μm filter), penicillin (100 U/mL) and streptomycin (100 ng/mL). To investigate the proteasomal degradation of our proteins of interest, electroporated cells were treated with MG-132 (Merck; New York, NY, USA), a proteasome inhibitor. MG-132 (final concentration 5 μM) was added to the cell culture media 12 h after transfection. The cells were collected after an incubation period of 8 h and analysed via Western blotting. DMSO (Sigma-Aldrich; St. Louis, MO, USA) was used as a negative control.

### 2.3. Western Blot and Flow Cytometry

MAGEA10 protein expression was detected in both cells and EVs using the Western blot method. The samples were lysed in Laemmli buffer and denatured at 100 °C for 10 min. Proteins from the cell and EV lysates were separated electrophoretically in 10 or 12% SDS-PAGE gel and then transferred onto a polyvinylidene fluoride (PVDF) membrane (Amersham^TM^; Global Life Sciences Solutions Operations UK Ltd., Buckinghamshire, UK) using the Trans-Blot SD Semi-Dry Transfer Cell (Bio-Rad Laboratories; Hercules, CA, USA). The membranes were incubated with primary antibodies overnight at 4 °C either with mouse monoclonal antibodies 5E11 detecting E2Tag (final concentration 0.1 ng/mL) (Icosagen Ltd., Tartu, Estonia); anti-α-tubulin, dilution 1:10,000, (T5168, Merck, St. Louis, MO, USA) or rabbit polyclonal antibodies anti-TSG101, dilution 1:10,000, (T5701, Merck, St. Louis, MO, USA), anti-MAGEA10 [26] or anti-EGFP, dilution 1:10,000 (Institute of Technology; University of Tartu, Tartu, Estonia). Secondary antibodies used were conjugated with HRP (goat anti-rabbit (0.8 mg/mL, Invitrogen; Waltham, MA, USA) and goat anti-mouse (0.8 mg/mL, Invitrogen; Waltham, MA, USA), dilution 1:10,000) and incubated for 45 min at room temperature. Proteins were visualized with ECL Western blotting reagents (GE Healthcare; Marlborough, MA, USA) according to the manufacturer’s protocol.

Flow cytometry analyses were conducted to analyse the expression levels of different MAGEA10 mutants fused with EGFP. Live Cop5-EBNA cells expressing MAGEA10-EGFP fusion proteins were collected 24 h post-transfection and suspended in PBS. GFP fluorescence was then measured with the Attune NxT Flow Cytometer (Thermo Fischer Scientific; Waltham, MA, USA).

### 2.4. Confocal Microscopy

For immunofluorescence analysis, Cop5-EBNA cells transfected with expression plasmids were grown on glass coverslips. Twenty-four hours after transfection, the cells were washed with PBS, fixed in 4% paraformaldehyde in PBS for 10 min at room temperature, and permeabilised with 0.2% Triton X-100 in PBS for 10 min on ice. The cells were blocked with PBS containing 5% bovine serum albumin (BSA) and incubated with primary antibodies overnight at 4 °C. Monoclonal antibody 1D12 recognizing the MAGEA10 protein (final concentration 5 µg/mL; Institute of Technology; University of Tartu, Tartu, Estonia) and a secondary antibody conjugated with Alexa-568 (Invitrogen; Waltham, MA, USA) at room temperature for 45 min were used. All of the antibodies were diluted in 3% BSA-PBS. The cells were washed with PBS, and SlowFade^®^ Gold antifade reagent with DAPI (Invitrogen, Waltham, MA, USA) was added. The samples were analysed with a confocal laser scanning microscope LSM710 (Carl Zeiss AG, Oberkochen, Baden-Württemberg, Germany), using the 63× objective. The images obtained were analysed with the ZEN2011 software (blue edition) (Carl Zeiss AG, Oberkochen, Baden-Württemberg, Germany).

### 2.5. Purification of EVs

EVs were isolated as described by Kowal et al. [29]. Forty-five mL of the cell culture media was collected 72 h post-transfection and centrifuged at 300× *g* for 10 min to remove dead cells and cell debris. The supernatant was then centrifuged at 2000× *g* for 20 min to precipitate apoptotic bodies and larger vesicles of similar size (large EVs). Next, the supernatant was centrifuged at 120,000× *g* using the Optima™ L-90K Ultracentrifuge, rotor SW28 (Beckman Coulter; Brea, CA, USA) at 27,000 rpm for 70 min to precipitate small EVs. All of the pellets were suspended in 200 μL of PBS. The samples were centrifuged again at 17,000× *g* for 15 min at 4 °C (large EVs) using the MicroCL 21R Centrifuge from Thermo Scientific (Waltham, MA, USA) or at 32,000 rpm for 1.5 h at 4 °C (small EVs) using the Optima™ L-90K Ultracentrifuge, rotor SW55 (Beckman Coulter; Brea, CA, USA). The washed pellets were suspended in 100 μL of PBS.

## 3. Results

### 3.1. The N-Terminal Intrinsically Disordered Region of MAGEA10 Is Responsible for Aberrant Migration in SDS-PAGE

In order to study the molecular characteristics of the MAGEA10 protein, we designed various MAGEA10 mutants, as depicted in Figure 1A,B. The N-terminal part (the first 120 amino acids; termed 1–120) and the C-terminal region of MAGEA10 containing the MHD (aa 121–369) were separated and linked to the epitope tag E2Tag or the EGFP protein. Next, we constructed plasmid encoding for the MHD domain alone (aa 121–327) and mutants 15–369 and 15–120, where the first 14 amino acids from the N-terminus of MAGEA10 were deleted. The N-terminus of MAGEA10 contains a nuclear localisation signal (NLS) similar to the SV40Large T antigen (aa 5 to 11) which is probably responsible for the nuclear localisation of MAGEA10 [24,25]. In all of the cases, the epitope tag, as well as the EGFP protein, were added to the C-terminus of polypeptides, leaving the N-terminus free (Figure 1A,B).

All of the MAGEA10 mutants were transfected into mouse fibroblast cells, and protein expression was analysed via Western blotting. As shown in Figure 1C, lane 2, the full-length MAGEA10 protein has an apparent molecular weight of 68–70 kDa, which differs significantly from its calculated MW of 42.3 kDa. The N-terminal 1–120 and C-terminal 121–369 mutants both migrated with an apparent molecular weight of 30–32 kDa (Figure 1C, lanes 4 and 7, respectively). This was expected for 121-369 (calculated molecular weight is 30 kDa), but not for 1-120, with a calculated MW of 14 kDa. Similar abnormal migration patterns were seen for EGFP-fused truncated proteins (Figure 1D, lanes 5, 8). The deletion of the first 14 amino acids drastically reduced the migration of the 15-369 protein, which moved in the gel as a 46 kDa polypeptide. However, the protein still exhibited a 5-kDa shift in comparison to the calculated molecular weight (Figure 1C, lane 3). The 15–120 polypeptide showed no expression when linked to the E2Tag (Figure 1C, lane 5), but a strong signal was detected when the protein was fused to EGFP (Figure 1D, lane 6). Again, the deletion of the first 14 amino acids reduced the aberrant migration-calculated molecular weights for 1–120-EGFP and 15–120-EGFP, which were 40.4 and 38.8 kDa, respectively; however, their mobility in the gel differed in units of more than 10 kDa (Figure 1D; compare lanes 5 and 6). The summary of the migration patterns of MAGEA10 deletion mutants is shown in Table 1.

In addition to aberrant migration, MAGEA10 mutants also showed different expression levels in mouse fibroblast cells. The expression of proteins in cells is controlled by the ubiquitin–proteasome pathway, which is an important protein quality control system in eukaryotic cells. The treatment of transfected cells with the proteasome inhibitor MG-132 enhanced the expression of 121–327 and 121–369 polypeptides containing MHD but did not have a significant effect on the expression of N-terminal 1–120 polypeptide (Figure 1E, lanes 4, 8, 10). The truncated N-terminus, 15–120-E2Tag, was not expressed either with or without proteasome inhibitor MG-132 (Figure 1E, lanes 5, 6). The quantification of Western blot images is shown in Appendix A. Next, we analysed the Mean Fluorescence Intensities (MFI-s) of EGFP signals of transfected cells via flow cytometry (Figure 1F). The 1–120-EGFP protein was detected at much higher levels than the C-terminal 121–369 or 121–327 proteins (Figure 1F). The deletion of the first 14 amino acids reduced the expression of the fusion proteins approximately two-fold, for both the full-length protein and the 1–120 mutant (Figure 1F).

Taken together, the N-terminal intrinsically disordered protein region (IDPR) of MAGEA10 is responsible for both the stabile expression and the aberrant migration of the protein in the gel. The C-terminal part of MAGEA10 containing the conserved MHD migrated in the gel similar to its calculated molecular weight; however, its expression level was decreased, probably being affected by the deletion of the N-terminal IDPR. Furthermore, both N- and C-terminal IDPRs stabilize the MHD region and protect it from degradation by proteasomes. In turn, the expression of the N-terminal IDPR, as well as the full-length protein, largely depended on the very first 14 amino acids, which had a great impact on the mobility of the polypeptides in SDS gel electrophoresis as well as on cellular expression.

### 3.2. Amino Acids K6 and R7 Are Required for the Nuclear Localisation of MAGEA10

Two adjacent positively charged amino acids in the putative NLS region, K6 and R7, are the most probable candidates responsible for localising the protein to the cell nucleus. To test this, we mutated K6 and R7 to alanines, both individually and together (Figure 2A). Western blot analysis confirmed the expression of all mutants in cells (Figure 2B,C). K6/R7 mutant was re-localised to the cytoplasm of the cell as detected via indirect immunofluorescence analysis using antibodies against MAGEA10 (Figure 2D). In some cells, the K6/R7A mutant was detected both in the cytoplasm and nucleus, suggesting that other regions of MAGEA10 might also contribute to the nuclear localisation of the protein.

### 3.3. The First Seven Amino Acids in the N-Terminus Are Responsible for Aberrant Migration

Deletion of the first 14 amino acids drastically reduced the migration of 15-369 MAGEA10 (Figure 1C, lane 3) and 15-120-EGFP mutants (Figure 1D, lane 6) in SDS-polyacrylamide gel. During NLS studies, we noticed that the MAGEA10 double mutant K6/R7 showed aberrant migration in the gel (Figure 2B, lane 5). Our next aim was to focus on the first 14 amino acids of MAGEA10 to study this phenomenon in more detail. First, we focused on the region 2-PRAPKR-7, mutated all the positively charged amino acids one by one to alanines (Figure 3A) and analysed the migration of the mutant proteins via Western blotting. As shown on Figure 3B, none of the single mutations (P2, R3, P5, K6, R7) affected the migration of the proteins in SDS-PAGE (lanes 3, 4, 6, 7, 8). The double mutants P2/R3A and K6/R7A, as well as the triple mutant P5/K6/R7A, caused a mobility shift in the gel (lanes 5, 9 and 10), suggesting that the proline residues preceding R3 and K6/R7 play a role in determining the structure of the MAGEA10 N-terminus. Mutant P5/K6/R7A exhibited two bands, one of which was similar to K6/R7A (lane 9) and the other moved approximately 7 kDa lower (lane 10). Mutating all the amino acids in motifs 2-PR-3 and 5-PKR-7 to alanines resulted in the polypeptide migrating as a 60-kDa protein in SDS-PAGE instead of the previously observed 68–70 kDa for wild-type MAGEA10 (lanes 2, 11). Inserting the first 14 amino acids of MAGEA10 in front of the EGFP protein did not affect the migration of the EGFP-fusion protein, which moved according to the calculated molecular weight (Figure 3C,D). Mutations of amino acids 2-PR-3 and 6-KR-7 to alanines within the 1-14 peptide fused to EGFP also did not affect the migration of the respective polypeptides (Figure 3C,D). Therefore, our data show that the first seven amino acids contained two linear motifs, 2-PR-3 and 5-PKR-7, which were involved in the mobility of full-length MAGEA10 in SDS-polyacrylamide gel; however, the N-terminal 14 amino acids did not function independently.

Next, we mutated linear motifs 2-PR-3, 6-KR-7, 5-PKR-7 and all of the positively charged amino acids within the region 2-PRAPKR-7 to alanines in the context of 1-120-EGFP, the N-terminal IDPR of MAGEA10 (Figure 3E). As shown in Figure 3F, all these mutated proteins moved aberrantly in the gel. Similar to full-length MAGEA10, mutant P5/K6/R7A exhibited two bands, the lower migrating 12 kDa below (lane 5). The replacing of all five positively charged amino acids with alanines resulted in the protein migrating approximately 10 kDa lower than 1-120-EGFP (Figure 3F, lane 6). The migration of P5/K6/R7A is comparable to that of the deletion mutant 15-120-EGFP (lanes 5 and 8), confirming that this linear motif is responsible for the aberrant movement in the gel. Mutating the negatively charged amino acids 13-EED-15 to alanines did not affect the migration of this polypeptide (lane 7). In summary, the positively charged amino acids within the region 2-PRAPKR-7 are responsible, at least partly, for its aberrant migration in SDS-polyacrylamide gel. This motif functions in the context of the N-terminal IDPR, encompassing amino acids 1-120, but does not cause aberrant migration when fused independently to EGFP.

### 3.4. Hiding the N-Terminus Affects MAGEA10 Expression and Mobility in Gel

In all the MAGEA10 mutants analysed so far, the EGFP protein was fused to the C-terminus of the protein, leaving the N-terminus free. At this point, we changed the position of EGFP, inserting it to the N-terminus of 1-120 of MAGEA10 (Figure 4A).

As a result, the expression of EGFP-1-120 was drastically decreased, and its mobility in the gel was similar to the P2/R3/P5/K6/R7A mutant, referred to as mut5 here (Figure 4B, compare lanes 2 and 4). There was no difference in apparent MW between EGFP-1-120 and EGFP-1-120mut5 (lanes 4 and 5). The change in expression level was confirmed via flow cytometry, showing a five-fold decrease for the constructs, where EGFP was located at the N-terminus (Figure 4C). The very low expression level of EGFP-1-120 and EGFP-1-120mut5 proteins was not caused by proteasome degradation, as the treatment with proteasome inhibitor MG-132 had no significant effect (Figure 4D). The quantification of Western blot images is shown in Appendix A.

Next, we inserted the small epitope tag E2Tag to the N-terminus of the full-length MAGEA10 to ensure that its expression also relies on the free N-terminus (Figure 5A). E2Tag is a small peptide (SSTSSDFRDR) and should not sterically hinder the structure of the protein. Again, the expression level of E2Tag-MAGEA10 was lower than that of MAGEA10-E2Tag, and its mobility in SDS-polyacrylamide gel was affected (Figure 5B, lanes 2 and 4). This time, E2Tag-MAGEA10mut5 (lane 5) moved differently from MAGEA10mut5-E2Tag, suggesting that other parts of the protein, in addition to the first seven amino acids, are also involved in the mobility of MAGEA10 in SDS-polyacrylamide gel. The treatment with proteasome inhibitor MG-132 increased the expression level of E2Tag-MAGEA10 and E2Tag-MAGEA10mut5 proteins; however, it still remained much lower than the expression of MAGEA10-E2Tag and MAGEA10mut5, respectively (Figure 5C, lanes 7–10). The quantification of Western blot images is shown in Appendix A. The location of the epitope tag did not affect the nuclear localisation of MAGEA10 (Figure 5D). Both wt proteins were localised in the nucleus, whereas, in the case of MAGEA10mut5-E2Tag and E2Tag-MAGEA10mut5 mutants, the nuclear localisation was disrupted, and the proteins were mainly detected in the cytoplasm.

In summary, the first seven amino acids, specifically the positively charged amino acid residues within the unique linear motif 2-PRAPKR-7 motif, are necessary for the expression and nuclear localisation of the protein. The same motif is also responsible for the aberrant migration of MAGEA10 in SDS-polyacrylamide gel; however, abnormal movement and nuclear localisation are not related to each other. Hiding the N-terminus with an epitope tag affected the protein’s mobility in gel and cellular expression. Therefore, our data show that the positively charged and rigid “head” of MAGEA10 is of significant importance in shaping its characteristics and activity in cells.

### 3.5. Mutations within the N-Terminus Does Not Affect MAGEA10 Incorporation into EVs

The MAGEA4 and 10 proteins are incorporated into nanosized biological vesicles, called extracellular vesicles (EVs), secreted into the conditioned cell culture media of transfected cells as well as COP5 cells expressing MAGEAs [25,26,30]. The relevance of this biological function of MAGEA proteins is not yet known, but we know that it is mediated by the C-terminal part of MAGEA4 containing the MHD region [25]. To test whether MAGEA10 mutant K6/R7 retained this function, we isolated EVs from the cell culture media of transfected cells (Figure 6A,B), using a two-step ultracentrifugation purification protocol to separate small EVs from large vesicles, and analysed them via Western blotting (Figure 6C). MAGEA10 incorporated into EVs of different sizes as shown in Figure 6C, lanes 5 and 8. The MAGEA10 K6/R7 mutant was also detected in both large and small EVs (Figure 6C, lanes 6 and 9). A similar result was obtained with mut5, where all the prolines, arginines and lysine within motif 2-PRAPKR-7 were mutated to alanines (Figure 6D, lanes 7 and 11). Furthermore, the N-terminal IDPR—amino acids 1-120—was not incorporated into EVs (Figure 6D, lanes 8 and 12), confirming that it is not required for this activity. TSG101, which is a cellular marker for exosomes, was detected only in small EVs, confirming the separation of small EVs from the large EVs consisting of apoptotic bodies and larger microvesicles. Thus, K6 and R7, within the proposed NLS, are required for MAGEA10 nuclear localisation, but the nuclear localisation itself has no role in protein incorporation into EVs.

## 4. Discussion

The MAGEA subfamily proteins have gathered interest as cancer biomarkers and targets for immunotherapy due to their immunological properties and distinctive expression patterns. MAGEA was firstly exploited to develop tumour vaccines; studies on protein functions commenced later. Given the significance of MAGEA10 in biology, and specifically in cancer, it is important to fully understand the functions of this protein at the molecular level in order to exploit it in translational research. In this current study, we demonstrate the significance of the N-terminus of MAGEA10 in its molecular functions. The N-terminus of MAGEA10 (i) is required for the expression of the protein in mammalian cells, (ii) is involved in the aberrant migration of the protein in SDS-polyacrylamide gel and (iii) contains a functional NLS. Blocking the N-terminus with an epitope tag or GFP drastically reduced MAGEA10 expression levels; however, its nuclear localisation was not affected, indicating that these functions are independent of each other. The free N-terminus is, at least partly, responsible for the aberrant migration of the protein in SDS-polyacrylamide gel electrophoresis, a phenomenon that has long been known [24]. We have identified the linear motif, 2-PRAPKR-7, responsible for the aberrant migration of MAGEA10 in SDS-PAGE. This linear motif is specific to MAGEA10, referring to the possibility that these characteristics are unique to this protein.

The N-terminus of MAGEA10 is an intrinsically disordered protein region (IDPR), responsible for the aberrant migration in SDS-PAGE. IDPRs are protein regions that lack rigid 3D structures either along their entire length or in localised regions. Despite the lack of structure, most IDPs can transition from disorder to order upon binding to biological targets and often promote highly promiscuous interactions. The majority of CTAs are IDPs or contain some part of the protein that is unstructured, according to bioinformatic predictions [17]. MAGEA10 domains showed a remarkably different expression in cells when separated from each other. The MHD alone was expressed at much lower levels than the full-length MAGEA10 and degraded by proteasomes, suggesting that both the N- and C-terminal IDPRs protected it from degradation. The N-terminal 1-120 IDPR was highly expressed and even enhanced the expression level of EGFP when fused to the N-terminus, showing expression-enhancing properties. While IDPRs are mostly known to negatively affect protein stability [31], there is evidence that they can also stabilise their binding partners, whether it is another protein [32] or a different region within the same protein [33]. Thus, it is possible that the N-terminal IDPR interacts with other parts of the MAGEA10 protein, stabilising them. Another possibility lies within the amino acid sequence of the N-terminal region. Fishbain et al. demonstrated that disordered protein tails with sequence bias protect the proteins from proteasomal degradation and that one such bias includes proline-rich regions [34]. The N-terminal 1-120 region is rich in proline residues (12.5%) compared to 121-369 (5.6%), and the first 14 N-terminal amino acids include three proline residues in close proximity to each other (21% prolines), suggesting that the unique biochemical characteristics of MAGEA10 are linked to its N-terminal proline-rich sequences. The computational analysis of order/disorder of 1-120 N-terminal residues in wild-type and mut5 was performed, but notable differences were not detected (Appendix A).

MAGEA10 is enriched in negatively charged amino acids (pI = 4); similar acidic proteins sometimes show aberrant migration in SDS-PAGE [35,36,37]. Greater-than-expected anomalies are observed for domains containing dense clusters of negatively charged residues and are explained by the fact that highly acidic domains electrostatically repel SDS [38]. However, the anomalous migration pattern of MAGEA10 is caused by the N-terminal linear motif PRAPKR, which is enriched in prolines and positively charged amino acids. Prolines provide exceptional conformational rigidity compared to other amino acids and may help to fix the structure of this fragment. This may help expose lysines and arginines, which are often post-translationally modified [39] and, in turn, have a significant impact on the structure and function of the protein. On the other hand, we cannot exclude the possibility that the positively charged rigid “head” is required for intramolecular interactions or specific interactions with other proteins or protein complexes that stabilise expression and/or cause anomalous migration in gel. None of the possibilities listed can be proven or disproved at this stage and require further research. The N-terminus of every protein is the first region to encounter the cellular environment and represents the first opportunity for the cell to steer a protein. Our results suggest that the N-terminal linear motif PRAPKR contains unique properties required for the biological role of MAGEA10.

All MAGEA subfamily members contain unstructured N- and C-terminal regions, which are not conserved and may determine the specific functions of the proteins. The sequence alignment of MAGEA proteins reveals varying levels of sequence identity. MAGEA10 and MAGEA11 share only 58% and 63% identity with other MAGEA proteins, and MAGEA5 is an outlier with a complete N-terminus but a partially deleted MHD, while MAGEA3 and MAGEA6 are nearly identical (96% identity) and more similar to MAGEA2 and MAGEA12 [40]. MAGEA proteins have been proposed to function as pro-oncogenes; however, the underlying mechanisms are sometimes poorly understood. The most studied members of the family are MAGEA3/6 and MAGEA11 proteins (for review see [41]). Some molecular functions of MAGEA proteins are conserved among family members; for instance, MAGEA2, 3/6 and 12 are shown to bind to the TRIM28 E3 ligase with similar strengths to boost ubiquitin ligase activity against p53 through a common MHD [15]. On the other hand, MAGEA11 recruits HUWE1 E3 ubiquitin ligase for modulating core oncogenic and tumour suppressor pathways [42]. MAGEA11 is a unique member of the subfamily, acting as a steroid hormone transcriptional coregulator, and it promotes mRNA alternative polyadenylation [42]. Much less is known about the molecular functions of other MAGEA subfamily members. Coleman et al. have studied all the MAGEA subfamily members under same conditions and show that MAGEA1, 3/6, 4, 8, 9, 11 and 12 increase the proliferation and anchorage-independent growth of HEK cells over-expressing these proteins, but MAGEA5 and 10 do not [40]. There are also differences in protecting cells from chemical stressors; for instance, MAGEA5- and 10-expressing cells are more viable with high doses of 5-Fluorouracil than others [40]. This suggests that the MAGEA10 protein is involved in cell transformation, unlike other family members.

Our study overall indicates that the N-terminal IDPR of MAGEA10 has unique properties that determine the expression of protein in cells. This distinguishes it from other proteins in this family and could be utilised for cancer-based drug design and cancer therapies.

## Figures and Tables

**Figure 1 biomolecules-13-01704-f001:**
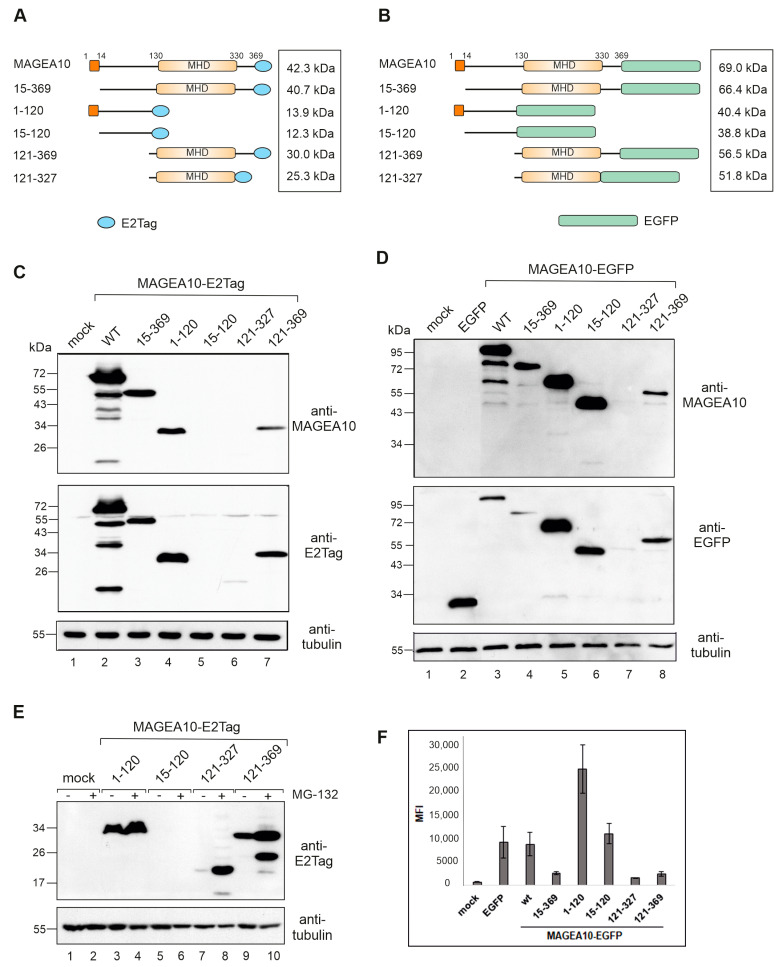
The N-terminal IDPR of MAGEA10 affects the electrophoretic mobility and expression of the protein in SDS-PAGE. (**A**,**B**) Schematic overview of the MAGEA10 deletion mutants fused with either E2Tag (**A**) or EGFP (**B**). The first 14 aa and the MHD are depicted as dark or light orange boxes, respectively. The C-terminal tags are depicted as blue ellipses (E2Tag) or green boxes (EGFP). The calculated molecular weight for each construct is shown on the right. (**C**,**D**) Western blot analyses of Cop5-EBNA cells expressing MAGEA10 deletion mutants fused with E2Tag (**C**) or EGFP (**D**). Signals were detected with antibodies against MAGEA10, E2Tag, EGFP and α-tubulin. (**E**) Western blot analyses of Cop5-EBNA cells expressing recombinant MAGEA10 proteins with and without MG-132 treatment. Original images can be found in Appendix A. (**F**) Flow cytometry analysis of Cop5-EBNA cells expressing recombinant MAGEA10 proteins fused with EGFP. Live cells were collected 24 h post-transfection and suspended in PBS, and EGFP-expressing cells were analysed. MFI—Mean Fluorescence Intensity.

**Figure 2 biomolecules-13-01704-f002:**
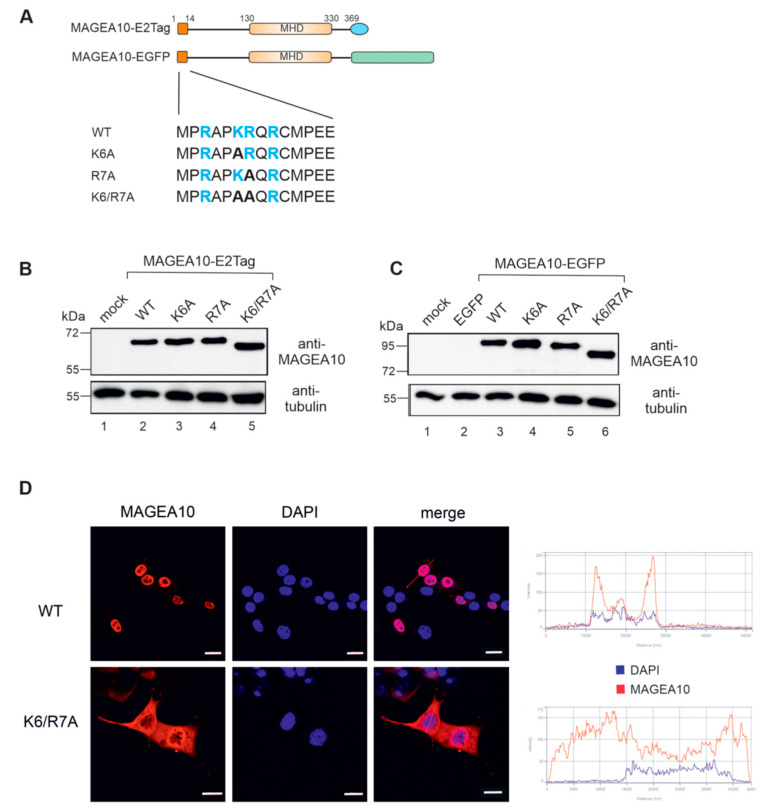
Amino acids K6 and R7 are required for the nuclear localisation of MAGEA10. (**A**) Schematic overview of the MAGEA10 K6/R7 point mutations. The first 14 aa and the MHD are depicted as dark or light orange boxes, respectively. The C-terminal tags are depicted as either a blue ellipse (E2Tag) or a green box (EGFP). The sequences of the wt and mutated N-termini are shown below. Positively charged amino acids are shown in blue. (**B**,**C**) Western blot analyses of Cop5-EBNA expressing recombinant MAGEA10 proteins. The signals were detected with antibodies against MAGEA10 and α-tubulin. Original images can be found in Appendix A; (**D**) Indirect immunofluorescence analysis of Cop5-EBNA cells expressing recombinant MAGEA10 proteins, using a MAGEA10-specific antibody, 1D12. Alexa-568 (Invitrogen)-conjugated antibody was used as a secondary antibody. DAPI was used to visualize cell nuclei. The fluorescence intensity of selected cells is shown on the right. Scale bar 20 μm.

**Figure 3 biomolecules-13-01704-f003:**
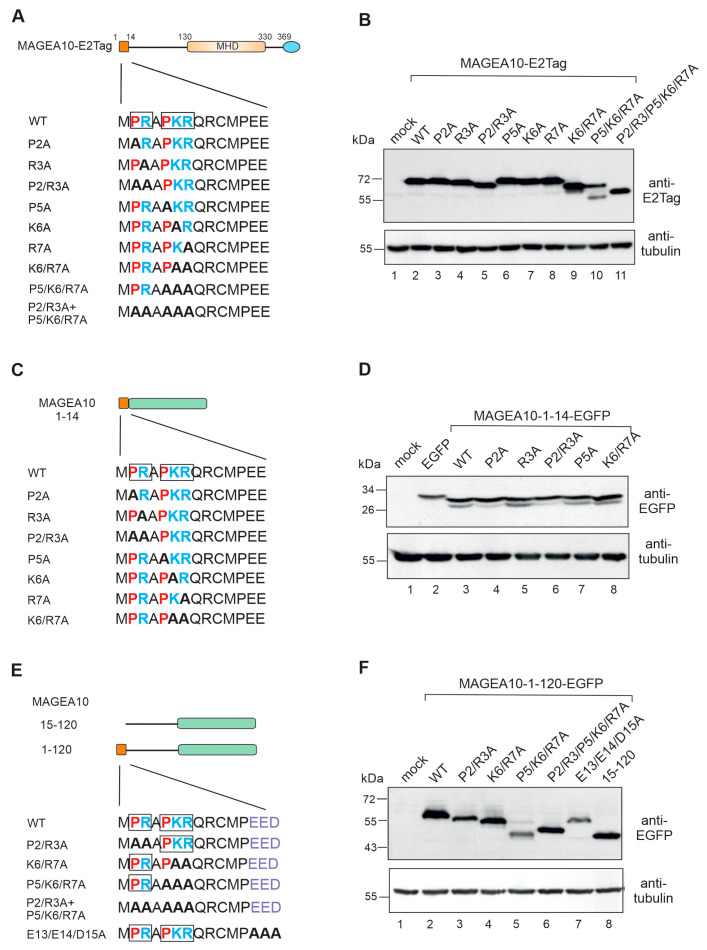
The first seven amino acids in the N-terminus are responsible for aberrant migration. (**A**) Schematic overview of the point mutations in region 2-PRAPKR-7. The first 14 aa are depicted as a dark orange box and the MHD are depicted as a light orange box. The C-terminal E2Ttag is depicted as a blue ellipse. The sequences of the wt and mutated N-termini are shown below. Positively charged amino acids are shown in blue, negatively charged in violet and prolines in red. (**B**) Western blot analysis of Cop5-EBNA cells expressing mutant MAGEA10 proteins. Signals were detected using antibodies against E2Tag and α-tubulin. (**C**) Schematic overview of point mutations within the first 14 aa of MAGEA10. (**D**) Western blot analysis of Cop5-EBNA cells expressing mutant MAGEA10 1-14 polypeptides fused with EGFP. Signals were detected using antibodies against EGFP and α-tubulin. (**E**) Schematic overview of deletion of and point mutations within the first 14 aa of MAGEA10 1-120 region. The sequences of the wt and mutated N-termini are shown below. (**F**) Western blot analysis of Cop5-EBNA cells expressing mutant MAGEA10 1-120 proteins. Signals were detected using antibodies against EGFP and α-tubulin. Original images can be found in Appendix A.

**Figure 4 biomolecules-13-01704-f004:**
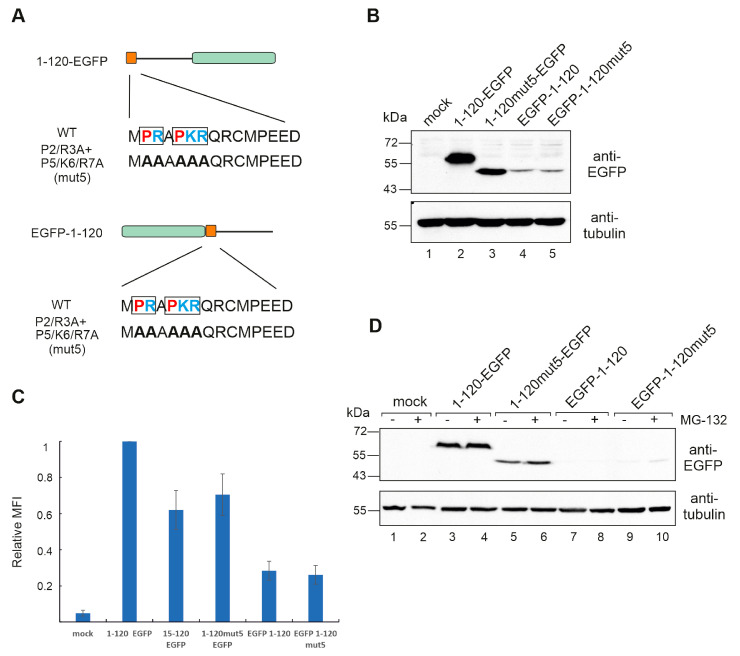
The linear motif 2-PRAPKR-7 in the N-terminal IDPR is responsible for aberrant migration. (**A**) Schematic overview of MAGEA10 1-120 wt and P2/R3A + P5/K6/R7A mutant (mut5) proteins. The first 14 aa are depicted as a dark orange box and EGFP is depicted as a green box. The sequences of the wt and mutated N-termini are shown below. Positively charged amino acids are shown in blue and prolines in red. (**B**) Western blot analysis of Cop5-EBNA cells expressing mutant MAGEA10 1-120 proteins. Signals were detected using antibodies against EGFP and α-tubulin. (**C**) Flow cytometry analysis of Cop5-EBNA cells expressing mutant MAGEA10 1-120 proteins fused with EGFP. Live cells were collected 24 h post-transfection and suspended in PBS, and EGFP-expressing cells were analysed. MFI—Mean Fluorescence Intensity. (**D**) Western blot analysis of Cop5-EBNA cells expressing mutant MAGEA10 1-120 proteins with and without MG-132 treatment. Signals were detected using antibodies against EGFP and α-tubulin. Original images can be found in Appendix A.

**Figure 5 biomolecules-13-01704-f005:**
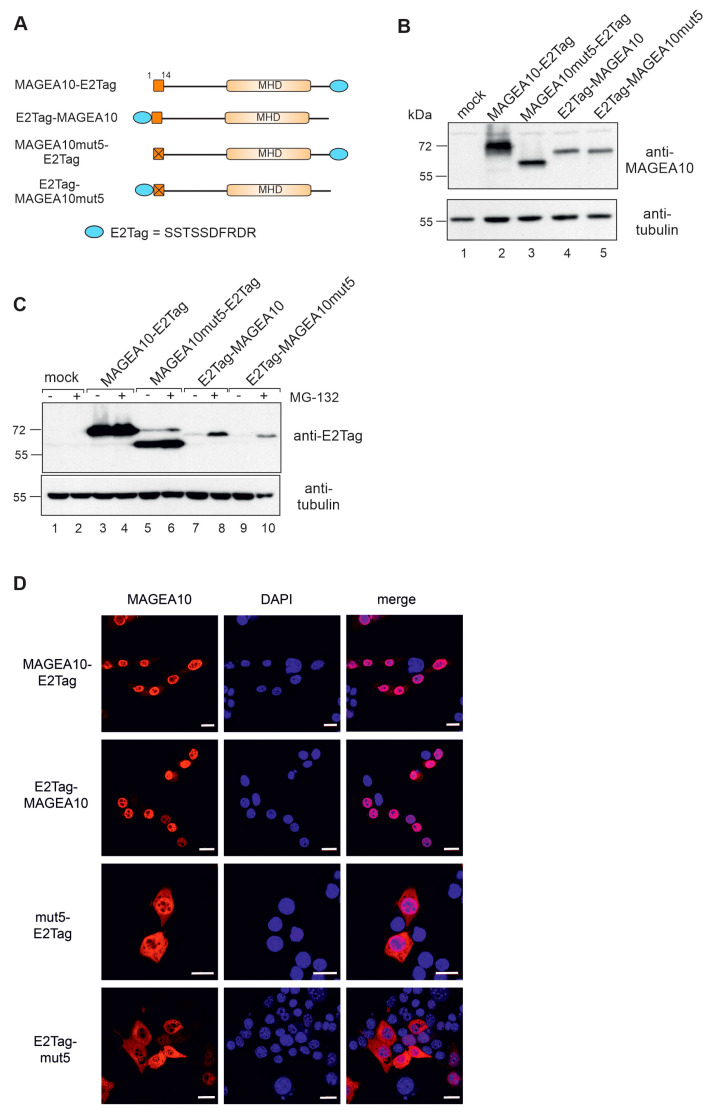
MAGEA10 expression requires a free N-terminus. (**A**) Schematic overview of wt and mut5 MAGEA10 proteins. The first 14 aa are depicted as a dark orange box, which has a cross inside it if there are mutations present. The MHD is depicted as a light orange box and the E2Tag is depicted as a blue ellipse. (**B**) Western blot analysis of Cop5-EBNA cells expressing mutant MAGEA10 proteins. Signals were detected using antibodies against MAGEA10 and α-tubulin. (**C**) Western blot analysis of Cop5-EBNA cells expressing MAGEA10 proteins with and without MG-132 treatment. Signals were detected using antibodies against EGFP and α-tubulin. Original images can be found in Appendix A (**D**) Indirect immunofluorescence analysis of Cop5-EBNA cells expressing MAGEA10 proteins, using a MAGEA10 specific antibody 1D12. Alexa-568-conjugated antibody was used as a secondary antibody. DAPI was used to visualize cell nuclei. Scale bar 20 μm.

**Figure 6 biomolecules-13-01704-f006:**
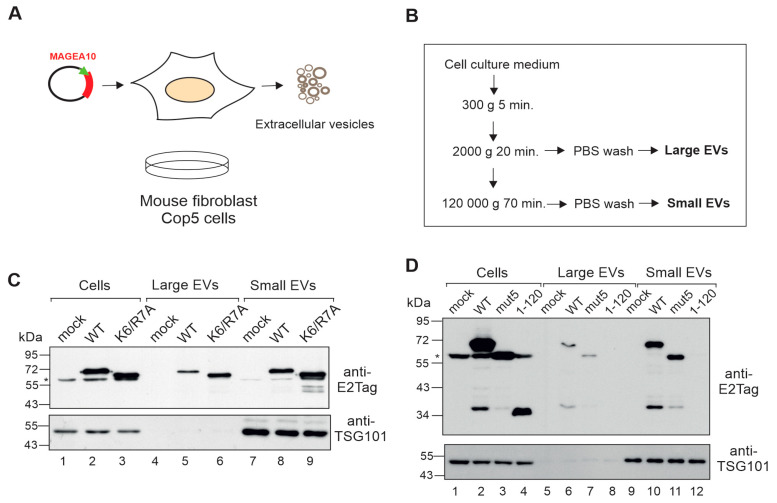
Mutations within the N-terminus of MAGEA10 do not affect MAGEA10 incorporation into EVs. (**A**,**B**) Schematic overview of the process of generating (**A**) and the subsequent purification of EVs (**B**). (**C**,**D**) Western blot analysis of Cop5-EBNA cells and EVs purified from Cop5-EBNA cells expressing recombinant MAGEA proteins. Signals were detected with antibodies against E2Tag and TSG101. The non-specific signal is shown by the asterisk (*). Original images can be found in Appendix A.

**Table 1 biomolecules-13-01704-t001:** Summary of MAGEA10 migration patterns in SDS-PAGE.

	Protein Length (aa)	Calculated MW (kDa)	Mobility in SDS-PAGE (kDa)	Times Higher in SDS-PAGE than Calculated
MAGEA10-E2Tag	383	42.3	68	1.6
15-369-E2Tag	370	40.7	46	1.1
1-120-E2Tag	133	13.9	30	2.2
121-327-E2Tag	221	25.3	25	1.0
121-369-E2Tag	264	30	32	1.1
MAGEA10-EGFP	620	69	98	1.4
15-369-EGFP	598	66.4	74	1.1
1-120-EGFP	369	40.4	60	1.5
15-120-EGFP	356	38.8	48	1.2
121-327-EGFP	457	51.8	52	1.0
121-369-EGFP	500	56.5	56	1.0

## Data Availability

The data presented in this study are available on the request from the corresponding author.

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
