# Peer review of "How the Intrinsically Disordered N-Terminus of Cancer/Testis Antigen MAGEA10 Is Responsible for Its Expression, Nuclear Localisation and Aberrant Migration"

_biomolecules, 2023, doi:10.3390/biom13121704_

Round 1

Reviewer 1 Report

Comments and Suggestions for Authors

To investigate the molecular basis of MAGEA10 expression and its anomalous mobility in gel, deletion and point-mutation analyses of the MAGEA10 protein were conducted. The authors found that the intrinsically disordered N-terminal domain, specifically the first seven amino acids containing a unique linear motif PRAPKR, is accountable for its expression, aberrant migration in SDS-PAGE, and nuclear localization. Notably, the aberrant migration in gel and nuclear localization were found to be independent of each other. Concealing the N-terminus with an epitope tag significantly impacted its mobility in gel and expression in cells. These results suggest that the intrinsically disordered domains flanking the MHD play a pivotal role in determining the distinctive properties of individual MAGEA proteins. But I have several following concerns:

1. The nucleic acid sequences (including gene names, regulatory sequences, and primer names) should be in italics.

2. Table 1 should use a standard three-line table.

3. Abbreviations should be defined when they first appear in the text. Such as "SDS-PAGE"...

4. Some numbers and quantifiers should have Spaces between them, please check the full text and modify according to the rules. Such as "5μM" in line 97, ...

5. Please compare the statistical differences between different WB results and different group treatments and mark them.

6. Please add a ruler to the microscope pictures in picture 2, and make a group of bright field maps and quantization maps according to fluorescence intensity for each group of pictures, so as to compare the differences in MAGEA10 nuclear localization among different treatments.

7. Please unify the format of references in the article, including the author's name, the case of words in the title of the article, the writing of the name of the journal, and the page number.

Comments on the Quality of English Language

Minor editing of English language required

Author Response

Thank you for reviewing the manuscript.

Here are answers to your concerns.

  1. The nucleic acid coding sequences were changed to italics as requested by the reviewer. Changes were made in the second paragraph of the Introduction section and in 2.1 Plasmids subsection in Materials and Methods.
  2. Table 1 was changed to standard three-line table as requested by the reviewer.
  3. We checked the manuscript and added abbreviations where they were missing.

Page 2: sodium dodecyl sulphate-polyacrylamide gel electrophoresis (SDS-PAGE) and extracellular vesicle (EV); page 3: polyvinylidene fluoride (PVDF); page 4: a nuclear localisation signal (NLS).

  1. We checked the manuscript and added spaces between numbers and quantifiers where they were missing.
  2. We quantified the western blot signals of Fig. 1C, D, E; Fig.4 B, D and Fig. 5B, C. These data are added to the western blot images shown in Supplementary materials, Figure S1. Western blot analysis is considered a semi-quantitate method and therefore we do not usually use it to obtain quantitative data. We have analysed the expression of EGFP-fusion proteins with flow cytometry (Fig. 1F and Fig. 4C) where the average expression of three different experiments (starting from electroporation of cells) is shown. The differences are statistically significant.
  3. We added rulers to the microscopic images in Fig. 2D and Fig. 5D. The quantification maps according to the fluorescence intensities of both WT and K6/R7 mutant were added to Fig. 2D. Unfortunately, we do not have bright field images for this experiment.
  4. We checked the references and unified the format.

Reviewer 2 Report

Comments and Suggestions for Authors

This manuscript is focused on the defining the role of N-terminal end of MAGEA10 protein. Through the truncation, mutagenesis and imaging studies the authors have showed, with the confidence, that N-terminus: i) is required for proper expression in mammalian cells, ii) is involved in the aberrant (the term which might need further exploration, at least in my perspective, see below), and iii) contains a functional nuclear localization signal (NLS, by the way, this abbreviation was not disclosed in the manuscript). The experimental design and data analysis is solid. 

My major problem with the manuscript is the lack of biochemical/biophysical characterization of their major finding – what exactly “aberrant” migration on SDS-gel mean? It is well known that IDP/IDPR could run up at the wrong (often, but not always, bigger) size on SDS-page, but this does not really tell us much (SDS electrophoresis is running under denatured conditions anyway). There could be some posttranslational modifications. With the mutational studies provided, the later is probably not the reason, but still is not entirely eliminated. The minimum the authors could do, is to provide computational analysis of order/disorder of the first 120 N-terminal residues and how mutations tested affect this order/disorder state (and this does not require any additional experiments). If they see a notable difference, it would be worth-while to express and purify the N-120 construct and its mutants and check whether predictions are valid using different method (such as CD, DLS, etc.).

To check how N-terminal IDPR can improve stability, its interaction with the rest of the protein (particularly MHD) can be studied (by ITC, for an example) and how listed mutations might affect the interaction in question.

To summarized, the is a solid, but not a complete, study of MAGEA10 and the functional role of its N-terminal IDPR.

Author Response

Thank you for reviewing the manuscript.

We agree with the reviewer that the “aberrant” migration in SDS-PAGE does not really tell us much, but there must be a reason for it. Often these are posttranslational modifications; for example glycosylation gives a relatively big shift in SDS-gel or ubiquitination (shift 8-9 kDa) that changes the mobility of proteins in gel under denatured conditions. We have considered these options but these variants are not consistent with our data. We have considered the possibility that the N-terminus interacts with MHD to improve the stability of the full-length protein and plan to make these experiments in the near future. But regardless of the abnormal migration, the exposure of the N-terminal linear motif affects significantly the protein expression and stability in cells. Given that this is a cancer-testis antigen often over-expressed in many cancers, it is important to study this phenomenon.

We made computational analysis of order/disorder of the first 120 N-terminal residues with three different programs AlphaFold, ADOPT and PrDOS. These data are added to the manuscript as Supplementary Figure S2 and lanes 424 – 426 in Discussion section. We did not see any major differences in the state of order/disorder when we compared 1-120 wild-type and mut5. The biggest difference was detected with AlphaFold where the existence of inner loop was detected in wt protein, but missing in mut5. It is very difficult and at this stage speculative to say what does this difference mean and does it have anything to do with the biological function of the protein. We are grateful to the reviewer for suggestions for further experiments.

Round 2

Reviewer 1 Report

Comments and Suggestions for Authors

The authors have addressed all my concerns. I recommend accepting it in current form.